# Teaching Geriatrics and Transitions of Care to Internal Medicine Resident Physicians

**DOI:** 10.3390/geriatrics5040072

**Published:** 2020-10-08

**Authors:** Shirley Wu, Nicholas Jackson, Spencer Larson, Katherine T. Ward

**Affiliations:** 1Section of Geriatrics, Division of General Internal Medicine, Department of Medicine, Harbor-UCLA Medical Center, Torrance, CA 90502, USA; kward@dhs.lacounty.gov; 2Lundquist Research Institute, Torrance, CA 90502, USA; 3Division of General Internal Medicine and Health Services Research, Department of Medicine, David Geffen School of Medicine at UCLA, Los Angeles, CA 90095, USA; njjackson@mednet.ucla.edu; 4Department of Medicine, Harbor-UCLA Medical Center, Torrance, CA 90502, USA; slarson@dhs.lacounty.gov

**Keywords:** graduate medical education, curriculum, transitions of care, safety net

## Abstract

(1) Background: Internal medicine (IM) resident physicians need to be trained to care for older adults and transition them safely across care settings. Objective: The study purpose was to evaluate the efficacy of a curriculum in geriatrics assessment and communication skills for transitions of care (TOCs) to IM resident physicians. (2) Methods: IM residents rotated for 4 weeks on the geriatrics consult service at a large public teaching hospital, where they received didactic lectures and clinical experience in consultation and transitional care. The curriculum was designed to meet consensus guidelines for minimum geriatrics competencies expected of IM residents. Previously validated and published assessment tools were used for geriatrics knowledge and attitudes. Locally developed tools were used to directly observe and rate communication skills, and self-assess geriatrics assessment and health literacy skills. The curriculum was evaluated using a quasi-experimental, nonrandomized, single-group pre- and post-test observational design. Data on 31 subjects were collected over 18 months and analyzed using mixed-effects models. (3) Results: Average knowledge scores improved from 65% to 74% (Δ9%, 95% CI 4–13%, *p* < 0.001). Communication skills improved by an average of 1.15 points (95% CI 0.66–1.64, *p* < 0.001) on a 9-point scale. Attitudes did not change significantly. Self-rated confidence in geriatrics assessment and health literacy skills improved modestly. (4) Conclusions: The curriculum is effective in teaching basic geriatrics knowledge and communication skills, and increasing self-confidence in geriatrics assessment skills. In settings where an inpatient geriatrics consult service is feasible, the curriculum may be a model for combining geriatrics and TOC training.

## 1. Introduction

### 1.1. Problem Identification and General Needs Assessment

Non-geriatricians need to be educated in minimum geriatrics competencies using evidence-based methods in order to adequately care for our aging population [1]. In 2013, there were approximately 3590 full time equivalent (FTE) geriatricians in the United States; based on projections, there will be a national shortage of 26,980 FTE geriatricians in 2025 [2]. Therefore, all internal medicine (IM) residents need to be trained in minimum geriatrics competencies to prepare them for caring for geriatrics patients in both inpatient and outpatient settings.

How to best train resident physicians in geriatrics is not clear. The Accreditation Council for Graduate Medical Education (ACGME) requires exposure to geriatrics for IM residents, but does not specify learning outcomes. Consensus guidelines for content in geriatrics skills, knowledge, and behaviors for internal medicine residents have been published; twenty-six minimum geriatrics competencies (MGCs) for IM/family medicine (FM) residents were developed in agreement with ACGME core clinical competencies [3]. However, most published geriatrics curricula address a limited number of MGCs [4], or are comprehensive programs requiring significant faculty development and additional funding [5,6,7,8]. Frequently reported outcomes assessments were Kirkpatrick levels 2a and 2b in knowledge and attitudes [4]. Residency programs should use training strategies that are outcomes based, directly assessing professional skills and behaviors, i.e., Kirkpatrick levels 3 and 4 [9,10,11,12,13].

### 1.2. Problem Statement

To prevent health care disparities where geriatricians are in shortage, IM resident physicians need to be trained to care for older adult patients and transition these high-risk patients safely across care settings. Geriatrics curricula need to specifically teach and assess documentable outcomes for geriatrics competencies in knowledge, attitudes, and communication skills, and demonstrate lasting retention.

### 1.3. Targeted Needs Assessment

Our internal medicine (IM) residency program trains 51 categorical residents annually and is primarily a single-site program at a 553-bed public teaching hospital. Our local needs were to develop a geriatrics rotation with a comprehensive curriculum in accordance with ACGME requirements, assess outcomes in knowledge and skills, and ultimately improve the care of older adult patients in our safety-net setting.

## 2. Materials and Methods

We developed the curriculum using Kern’s six steps for developing medical education curricula [14], and evaluated the curriculum using a quasi-experimental, nonrandomized, single-group pre- and post-test observational design.

### 2.1. Curriculum Conceptual Framework

We selected content topics related to MGCs based on the care needs and characteristics of our safety-net hospital setting (see Table 1). Twenty-four of the 26 MGCs were mapped to specific learning materials, using educational principles of adult learning—in particular, deliberate practice, formative feedback, and scaffolding (Table A1). The curriculum provides a structure for participants to focus on practicing specific skills and receive constructive feedback through direct observation [14] (Appendix B). Peer-reviewed educational materials and assessment tools were used. Evaluation and feedback methods were chosen using the principles of Miller’s framework for clinical assessment [15]. Learning materials were available to the residents for self-study on an electronic share drive.

### 2.2. Rotation Format

The educational intervention was a 4 week clinical rotation in geriatrics and TOC. The curriculum consisted of a lecture series, direct patient care, and direct observation of clinical performance, with focused, formative feedback.

(1)There were 30 min lectures given by the geriatrics faculty (board certified) on thirteen geriatrics topics (Table 1). Peer-reviewed lectures were obtained from the Portal of Geriatrics Online Education (POGOe) [16], the American Geriatrics Society, or were locally developed.(2)Direct patient care on the inpatient geriatrics consult service, which targeted a high-risk, vulnerable elderly population with low health literacy and socioeconomic status, included:
Conducting a comprehensive geriatrics assessment (using the assessment packet),Performing a health literacy assessment,Educating patients and caregivers on medications and chronic disease management using the Personal Health Record tool,Daily patient assessment and presentation on attending rounds,Documenting communication with the PCP/outside provider, andEvaluating patients after discharge in the Transitions of Care Discharge Clinic.Preoperative frailty assessment in geriatrics patients has gained traction as a predictor of post-operative complications and therefore possibly an important modifier to guide care [17,18,19]. Therefore, patients aged 70 and above admitted to our hospital’s surgical services who met frailty screening criteria were automatically evaluated by the geriatrics consult service. Patients with dementia, depression, stroke, fall, hip fracture, or readmission within 90 days triggered the automatic consult. Our care transitions program is based on the Coleman model, which incorporates a Transitions Coach to provide continuity of care between inpatient and outpatient, targeted patient education, and medication reconciliation [20].(3)Direct observation and feedback by geriatrics attending physicians on residents’ clinical performance of geriatrics assessment tools and communication skills. Residents were provided a skills checklist to track practice of each skill (Appendix A: Geriatrics Skills Checklist) and facilitate formative feedback. Residents were observed during patient encounters and given focused, formative feedback by the attending physicians using a locally developed Geriatrics Communication Skills Mini-CEX, which comprises specific items that were scored by the attending physicians (Clinical Evaluation Exercise) (Appendix A: Geriatrics Communication Skills Mini-CEX).

### 2.3. Study Design

To meet ACGME requirements, the 4 week clinical rotation in geriatrics was required for our IM residents (PGY2 and PGY3). All IM residents on the rotation (*n* = 52) were recruited from November 2014 through June 2016. Data were collected and managed using the REDCap electronic data capture tools hosted at the Lundquist Research Institute (Los Angeles BioMedical Research Institute) [21]. The local Institutional Review Board reviewed and approved the study protocol as exempt (LA BioMed IRB # 30248-01). Participants were assigned study identification numbers, and the data were de-identified for analysis by the statistician.

### 2.4. Evaluation

The curriculum was evaluated using a quasi-experimental, nonrandomized, single-group pre- and post-test observational design (Table 2).

Geriatrics clinic knowledge, attitudes towards geriatrics patients and clinical care, communication skills, and reaction to the curriculum were assessed pre- (O1) and post- (O2) rotation. Residents completed the University of Michigan Geriatrics Clinical Decision-Making Assessment (UM Geriatrics Assessment), a 21 item multiple-choice test that was used to assess knowledge outcomes, which was scored by the study investigators [22]; the University of California, Los Angeles (UCLA) Attitudes Scale (self-assessment comprising 14 items using a Likert scale) to assess attitudinal outcomes [23]; and a retrospective pre-post survey on knowledge and attitudes on health literacy (self-assessment comprising 11 items using a Likert scale). The attendings completed the Mini-CEX on communication skills (10 items using a Likert scale), which were administered at O1 and O2. Knowledge retention was assessed six to twelve months after the completion of the rotation (O3) by repeating the UM Geriatrics Assessment. The investigators developed a Program Satisfaction Survey (available in Appendix A) that was administered at O2 to the resident participants to assess overall curriculum effectiveness.

### 2.5. Statistical Analysis

Changes in clinical knowledge over time (O1, O2, and O3) were assessed using a generalized linear mixed model with binomial family and logit link function specified. These models were fit with person random intercept and nested random slope for time. Responses to each of the 21 clinical knowledge items were scored as correct or incorrect and global scores for the proportion correct across all items at each time point were estimated using the aforementioned mixed model. Specifically, the participants’ 21 binary items were modeled as the outcome with a fixed and random effect for time. The marginal means from this model were used to represent the overall knowledge scores at each time point. For the UCLA attitudes items, communication skills, and health literacy assessments, Cronbach’s alpha was used as a measure of scale reliability. Using the formed scales, a mixed-effects linear regression model was used to assess changes over time in a manner similar to those described for the knowledge assessments. Differences in baseline values and residency year were assessed using *t*-tests and Fischer’s exact tests between those with and without complete data at all timepoints. All analyses were conducted in Stata version 15.1, Stata Corp LP (College Station, TX, USA).

## 3. Results

Of the 52 participants, 48 provided a pre-test assessment (O1), 41 completed the first post-test assessment (O2), and 24 provided for the second post-test assessment at 6–12 months (O3) after rotation. There were no statistically significant differences between those with and without missing data on residency year (PGY), the baseline values of knowledge scores, observed communication skill, or self-rated geriatric assessment and health literacy skills.

Knowledge, observed communication skills, and self-rated geriatrics assessment and health literacy skills improved. Attitudes towards geriatrics and geriatrics patients did not change significantly.

Average knowledge scores improved from 65% (95% CI 62–68%, *p* < 0.001) to 73% (95% CI 70–76%, *p* < 0.001) (Figure 1). Forty-six percent (24 of total *N* = 52) completed the knowledge assessment at O3, with average scores of 74% (0.70, 0.78); an undetectable difference from the average scores at O2.

Communication skills improved by an average of 0.93 points (95% CI 0.54–1.33, *p* < 0.001) on a 9-point scale (Figure 2). Self-rated confidence in geriatrics assessment skills improved by an average of 1.32 points (95% CI 1.18–1.47, *p* < 0.001) on a 5-point scale, and health literacy skills improved by an average of 1.42 points (95% CI 1.22–1.63, *p* < 0.001) on a 5-point scale (Figure 2).

Analyses for interactions among variables did not show that improvement in knowledge was associated with improvement in attitudes, communication skills, or health literacy.

The Mini-CEX generated specific attending feedback to residents based on direct observation of clinical skills. The comments addressed communication skills (e.g., “talk directly to patient (not translator),” and “ask about pain and mood at beginning—acknowledge emotion”), using literacy-appropriate language and avoiding medical jargon (e.g., “‘confounding’, ‘mental status’, ‘Foley’—examples of jargon used”), and demonstrating active listening and empathy (e.g., “patient started to cry; did not acknowledge emotion”).

Participants self-rated their knowledge and skills in health literacy, transitional care, and geriatrics assessment as improved after the rotation.

Participants demonstrated increased awareness of limited health literacy in older adults, effective methods for screening for low health literacy, an increased awareness of the added health costs associated with low health literacy, and an increased preparedness to use “teach back” methods for communication.

The Program Satisfaction Survey assessed the learners’ reaction to the curriculum. Learners showed improved confidence in their ability to communicate effectively with outpatient providers about patient’s hospitalization and recommended follow up. Comments suggested that participants used deliberate practice to improve confidence in clinical skills (e.g., “I had not used many of the aspects of the geriatrics exam regularly. Now after performing the exam multiple time[s] I feel more comfortable and aware of those functional tests”) and had developed confidence in translating skills across care settings (e.g., “I feel much more confident with the geriatrics assessment and feel that it is a tool I can use on a lot of my clinic patients”).

In summary, participants’ geriatrics clinical knowledge, communication skills rated by faculty, and learners’ self-assessment of their geriatrics clinical skills improved after the curriculum. Knowledge was retained after six months.

## 4. Discussion

For the primary outcome of geriatrics knowledge scores, the curriculum was as effective at teaching basic geriatrics knowledge to residents as previously published geriatrics curricula, with similar improvements in knowledge scores. The lack of improvement in attitudes towards geriatrics patients and clinical care may be due to the short time frame of the educational intervention or biases against older adults developed earlier in life.

The curriculum meets current trends in medical education by teaching knowledge and skills for preparing residents for practice in new models of care since the Affordable Care Act. The curriculum provides a means to document direct assessment of clinical performance, such as in practice-based learning and improvement, interpersonal communication skills, professionalism, and IM Entrustable Professional Activities (EPA), e.g., managing transitions of care, modeling cross-cultural communication, and establishing therapeutic relationships with persons of diverse socioeconomic backgrounds.

The curriculum is potentially reproducible at other institutions; most of the curricular materials are publicly available on POGOe [16].

Future directions include revision and validation of the locally developed Geriatrics Communication Skills Mini-CEX, and developing a template for reportable competency outcomes relevant to residency program leadership. Assessing patient-level outcomes for patients evaluated on the geriatrics consult service would evaluate the educational impact on quality of patient care, particularly in a safety-net population.

### Limitations

This curriculum was administered at a single site and the residents and patient population may not be representative of the target population of internal medicine residents training in all safety-net settings. The patient population represented in our geriatrics consult service is enriched in surgical patients due to an agreement with the surgical leadership at our institution to perform geriatrics assessment on all geriatrics surgical patients meeting pre-specified frailty screening criteria.

While patient goals of care (GOCs) and caregiver capability and strain are important components in geriatrics assessment and transitional care, formal GOC training and caregiver assessment tools were not included in the curriculum. At our institution, the IM residents have a required rotation in palliative care, which includes curricula in how to elicit and document goals of care. If other institutions do not have a similar service or curricula, these elements of the curricula should be made more robust to help residents develop these essential skills.

The ABIM Mini-CEX tool has been previously shown to require a minimum of four observations to have sufficient reliability for summative assessment of clinical skills [24,25]. While there was an improvement in scores on the Mini-CEX in this study, this finding should be interpreted cautiously. The intended utility of the Mini-CEX in this curriculum was primarily for facilitating and documenting formative feedback.

Primary outcomes at Kirkpatrick level 4, such as residents’ quality of clinical care in practice after the rotation (e.g., rates of screening for dementia) and patient-level outcomes (e.g., length of stay, 30 day hospital readmission rates) were not assessed.

## 5. Conclusions

The curriculum was effective in teaching basic geriatrics knowledge and communication skills, and increased self-confidence in geriatrics assessment skills. Knowledge was retained at between six and twelve months. Attitudes towards geriatrics patients remained unchanged. Most of the didactic and assessment tools are publicly available. The curriculum may be a model for combining geriatrics and TOC training in safety-net hospital settings.

## Figures and Tables

**Figure 1 geriatrics-05-00072-f001:**
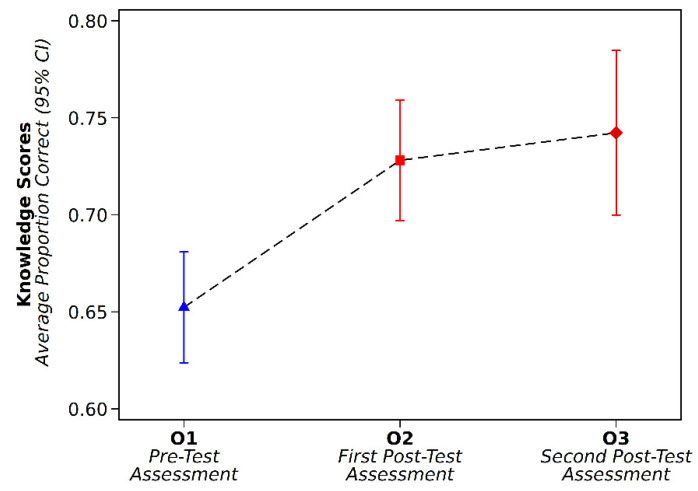
Change in Knowledge Scores.

**Figure 2 geriatrics-05-00072-f002:**
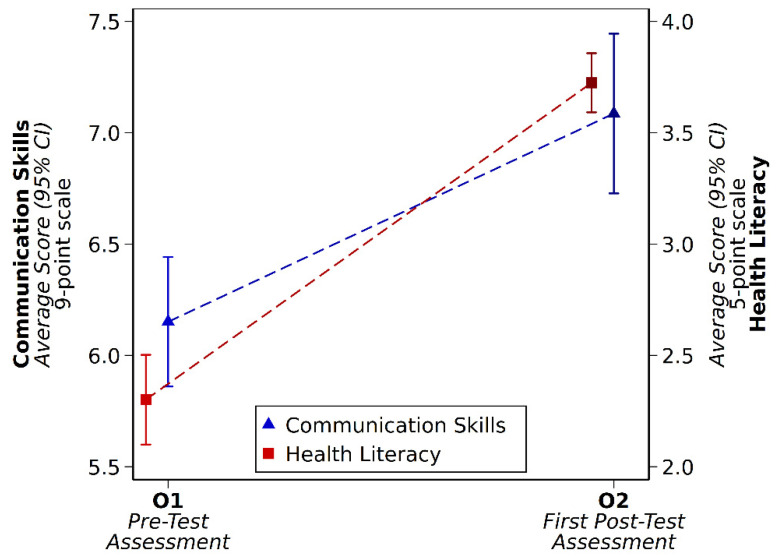
Change in Communication Skills and Health Literacy Skills.

**Table 1 geriatrics-05-00072-t001:** Geriatrics Content Topics.

1	Transitions of Care
2	Health Literacy and Health Disparities
3	Interdisciplinary Teams
4	Pre-Op/Peri-Operative Care
5	Pressure Ulcers
6	Urinary Incontinence
7	Delirium
8	Dementia
9	Osteoporosis and Hip Fractures
10	Gait Disorders and Falls
11	Appropriate Medications and Polypharmacy
12	Anticoagulation
13	Geriatrics Primary Care and Screening

**Table 2 geriatrics-05-00072-t002:** Learner Assessments.

Assessment	Instrument	O1	O2	O3
Knowledge	University of Michigan Geriatrics Clinical Decision-Making Knowledge Assessment (21 items)	x	x	x
Attitudes	UCLA Geriatrics Attitudes Survey and Carolina Geriatrics Education Center Health Literacy Survey	x	x	
Communication	Locally Developed Mini-Clinical Evaluation Exercise (Mini-CEX)	x	x	
Skills				
Curriculum	Program Satisfaction Survey		x	
Effectiveness				

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
