# Peer review of "Teaching Geriatrics and Transitions of Care to Internal Medicine Resident Physicians"

_geriatrics, 2020, doi:10.3390/geriatrics5040072_

Round 1
Reviewer 1 Report
Strong report on the major and pervasive need to expand skills in geriatrics and in effective transitional care. Contributes important advances in an approach to improve skills in geriatrics care and transitions of care to non-geriatrics clinicians
Abstract: Consider clarifying that the geriatric training included both didactic topical lectures and clinical experience in geriatric consultation. Such as:
L17-18 modify to "(2) Methods: IM residents rotated for 4 weeks 17 on the Geriatrics Consult Service at a large public teaching hospital, receiving topical didactive lectures and clinical experience in consultation and transitional care."
Important elements of both geriatric care and effective transitions of care include eliciting and documenting patient goals of care, as well as caregiver assessment for capability, strain, and need for supportive services. Suggest making these elements of care more evident in the main text if they were addressed, or if not, identifying them as gaps or limitations in the discussion. For example, while Box 1 mentions, "functional goals" and "personal values," a strong goals of care conversation goes beyond "functional goals" of restoring function and "personal values," eliciting and documenting personal goals of care. Clarification of coverage may fit in Line 71. While line 97 mentions caregivers, I did not see any mention of assessing caregiver strain nor providing needed caregiver support.
An important indicator of successful transitional care a reduction in 30 day hospital readmission fates. Was data collected that could feasibly report on 30 day readmission rates, perhaps providing results at time of O2 post evaluation or at time of O3 six to nine month evaluation (lines 142 or 143). If not, might this be considered as a limitation, mentioned in line 237.
Line 102: Pleased to see the connection and description of preoperative surgical frailty assessment, the trigger for geriatric consultation, and the experience and insights gained by residents. It would be appropriate to include a reference to this work, perhaps citing T. Robinson, J. Johanning, E. Finlayson, or D. Hall.
References - a few appear to be incomplete, such as 10 and 33.
Reviewer 2 Report
Overall this is a very well done manuscript that addressed the important issue of internal medicine resident training in geriatrics and transitions of care. It has sound methodology and the reporting of methods and results are clear. Not all sites have comparable clinical rotations, however the non-direct patient care activities all appear to be easily reproducible in other sites with minimal adaptations.
I only have minor feedback on the manuscript as below:
Introduction: Overall very well done with an excellent description of the current state of education and needs.
Line 38: I believe there is more recent data on the numbers of geriatricians than 2012. Consider updating your numbers/reference.
Materials and Methods:
Table 1: Is it possible to cite the origin of each lecture? It would be helpful for readers to know which lectures they can get from POGOe, and how to access them. Alternatively, a footnote could be added to direct readers to the table at the end of your manuscript where further information can be found.
Appreciate the supplemental materials supplied to help with reproducibility
There is a discrepancy in the description of the UM Geriatrics Assessment. Table 2 states it is 30 items and in the manuscript it states it is 21 items.
Results:
No suggested changes
Discussion:
The methods discussed mapping the MCG to specific learning materials, but did not mention mapping to the ACGME milestones. However, the discussion mentions mapping to the milestones. Can this be clarified in either the methods section, the discussion section, or both?
IM Resident Geriatrics Competencies & Didactic Materials Table:
If the ACGME milestones were mapped to the competencies and didactic materials, please include them in this table.
The delirium topic competency #6 does not have associated learning materials or assessment. Were there materials or assessments associated with this competency?
